# Training Fair Tabular Foundation Models

Patrik Kenfack [1 2]   Jesse C. Cresswell [3]   Anthony L. Caterini [3]   Samira Ebrahimi Kahou [4]   Ulrich Aïvodji [1 2]

## Abstract

Tabular Foundation Models (TFMs) have emerged as leading methods for tabular predictive tasks, leveraging in-context learning to predict on new data without task-specific training. Despite the increased use of TFMs in high-stakes decision-making, their fairness properties remain largely unexplored. In this work, we incorporate fairness constraints directly into TFM training, enabling fair predictions in a single forward pass. Our approach addresses two key challenges: limited access to sensitive attributes in training data, and the incompatibility of existing fairness techniques with the in-context learning paradigm. We propose FairTFM, a scalable training strategy based on synthetic fairness tasks and a fairness-aware architecture using a gradient reversal layer, which encourages the model to learn representations invariant to sensitive attributes. Experiments on 120 fairness tasks show consistent improvements in fairness while maintaining competitive accuracy.

## 1. Introduction

Tabular data, organized in rows and columns, is the dominant modality for decision-making tasks in domains such as healthcare and finance. While tree-based models, such as XGBoost (Chen & Guestrin, 2016), have long dominated tabular learning, tabular foundation models (TFMs) have emerged as strong alternatives (Hollmann et al., 2023). These models are typically pretrained on large collections of tabular data and can adapt to new tasks via in-context learning (ICL) (Brown et al., 2020), using only a few labeled examples. Unlike traditional approaches, TFMs do not require task-specific training or hyperparameter tuning. Notably, models such as TabDPT (Ma et al., 2025), TabPFNv2 (Hollmann et al., 2025), and TabICLv2 (Qu et al., 2026) can match or outperform heavily tuned tree-based models across a wide range of datasets (Erickson et al., 2021).

[1]ÉTS Montréal [2]Mila - Quebec AI Institute [3]Layer 6 AI, Toronto, Canada [4]University of Calgary. Correspondence to: Patrik Kenfack <patrik-joslin.kenfack.1@ens.etsmtl.ca>, Jesse C. Cresswell <jesse@layer6.ai>.

*Preprint. June 30, 2026.*

Even as TFMs are deployed in high-stakes decision-making, their fairness properties remain largely unexplored. Recent work shows that, despite strong predictive performance, TFMs can exhibit biased outcomes similar to traditional models (Kenfack et al., 2026). While fairness-aware training has been extensively studied in classical machine learning (Mehrabi et al., 2021), these approaches do not readily extend to the ICL paradigm, where predictions must be produced in a single forward pass without task-specific optimization. This gap calls for fairness-aware training methods specifically designed for TFMs. Hence, in this work, we propose a training framework that incorporates fairness as a first-class objective in TFMs alongside predictive performance. We train TFMs under explicit fairness constraints, enabling fair predictions directly through ICL without requiring post-hoc correction or task-specific retraining.

Our method relies on two key components. (i) **Synthetic fairness task generation:** given a dataset we randomly designate an input feature as a sensitive attribute, treat it as categorical, and optimize for fairness with respect to it. Repeating this process for every dataset sampled during pretraining enables scalable fairness-aware training across diverse tasks. (ii) **Fairness-aware architecture and training:** we extend transformer-based TFMs with a dedicated encoder for sensitive attributes and introduce a dual-head prediction mechanism. In addition to the label prediction head, we include a sensitive attribute predictor connected through a gradient reversal layer (Ganin et al., 2016), encouraging the model to learn representations invariant to sensitive attributes.

We evaluate our approach on 120 fairness tasks derived from the ACS PUMS dataset (Ding et al., 2021) with varying sensitive attributes. We measure fairness using demographic parity, equal opportunity, and equalized odds and compare against strong baselines. Our results show that the proposed FairTFM framework consistently improves fairness metrics while maintaining competitive predictive performance.

Our contributions are as follows:

- We propose a scalable strategy for creating fairness tasks with a wide variety of sensitive attribute relationships.
- We design a fairness-aware transformer architecture using adversarial learning that enables fair predictions in a single forward pass.

- We provide an extensive empirical evaluation on a diverse set of tasks, demonstrating improved fairness metrics while preserving accuracy.

## 2. Related Work

**Fairness** Prior work on fair machine learning typically intervenes at one of three stages: preprocessing the data, incorporating fairness constraints during training, or post-processing model outputs (Kamiran & Calders, 2009; Zemel et al., 2013; Hardt et al., 2016; Zhang et al., 2018). These approaches have been studied extensively for conventional supervised models, but they often assume access to the training pipeline or to calibrated model outputs. This assumption is less compatible with ICL, where a pretrained model is used as a frozen predictor at inference time. Our setting is therefore closer to learning fairness-aware representations during pretraining so that fair behavior can be obtained in a single forward pass.

**Tabular Foundation Models** Recent tabular foundation models such as TabPFN, TabDPT, and TabICL demonstrate that pretrained transformers (Vaswani et al., 2017) can be highly competitive on tabular prediction tasks (Hollmann et al., 2023; 2025; Ma et al., 2025; Grinsztajn et al., 2026; Qu et al., 2026). However, this literature emphasizes predictive accuracy, and does not directly address the potential for biased outcomes. FairPFN (Robertson et al., 2025) does incorporate fairness into TFM pretraining under a causal notion of fairness, which contrasts with our work which targets statistical group fairness notions such as demographic parity, equal opportunity, and equalized odds. Statistical fairness is data-driven and focuses on distributional fairness (outcome-based), while causal fairness is more interventional and does not necessarily ensure outcome parity across groups. We provide a more detailed discussion of these connections in Appendix A.

## 3. Fair Tabular Foundation Models

In this section, we present our fairness-aware tabular foundation model for statistical fairness.

### 3.1. Fairness Task Sampling From a Data Prior

Most TFMs are pretrained on tasks sampled from a data prior, whether synthetic or representing real datasets, in order to expose the model to a broad distribution of supervised learning problems. We follow the same principle, but augment it to generate fairness tasks in a self-supervised manner that capture diverse forms of group-dependent bias. Concretely, starting from a sampled dataset, we randomly designate one input feature as the sensitive attribute, as illustrated in Figure 1a. A similar process was employed in TabDPT (Ma et al., 2025) for crafting diverse predic-

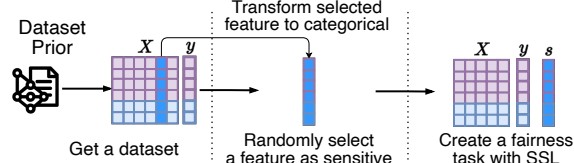

*(a)* Fairness task sampling from a prior.

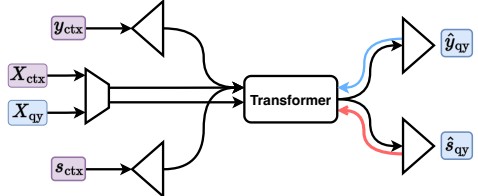

*(b)* Overview of the FairTFM architecture.

*Figure 1.* (a) We construct fairness tasks from a prior data generator by first sampling a dataset $(X, y)$, then randomly selecting one feature to serve as the sensitive attribute $s$ and converting it to a categorical variable, yielding the triplet $(X, y, s)$. The resulting data are split into context $(X_{\mathrm{ctx}}, y_{\mathrm{ctx}}, s_{\mathrm{ctx}})$ and query $(X_{\mathrm{qy}}, y_{\mathrm{qy}}, s_{\mathrm{qy}})$ sets for ICL. (b) The model jointly processes context and query inputs with a transformer. In addition to the main prediction head for the target label $\hat{y}_{\mathrm{qy}}$, a second head predicts the sensitive attribute $\hat{s}_{\mathrm{qy}}$. A gradient reversal layer on the sensitive branch encourages the model to learn representations that are invariant to $s$, thereby reducing reliance on sensitive-correlated features while preserving predictive performance.

tive tasks from real-world datasets, and in CausalPFN (Balazadeh et al., 2025; Stith et al., 2026) for generating causal inference tasks.

When the selected feature is continuous, we convert it into a categorical attribute using a "stick-breaking" discretization scheme. We first sample a sequence of proportions from Beta distributions and use them to construct mixture weights via a Dirichlet process (Khan et al., 2012), where each component represents a fraction of the remaining mass. These weights are then mapped to empirical quantiles of the normalized feature, producing data-dependent cut points that partition the feature into discrete groups. This procedure yields flexible, non-uniform bins that adapt to the underlying feature distribution while retaining a probabilistic interpretation.

### 3.2. Network Architecture

We build on TabPFN's transformer encoder for tabular data, using the nanoTabPFN (Pfefferle et al., 2025) architecture as a lightweight backbone that also has less input pre-processing so that we mitigate confounding effects. In this architecture, each input pair $(X_i, y_i)$ is represented as a sequence of $d$-dimensional tokens and processed by alternating self-attention over rows and columns, enabling ICL along both axes. As in TabPFN, the query label $y_i$ is masked during both training and inference, and the corresponding

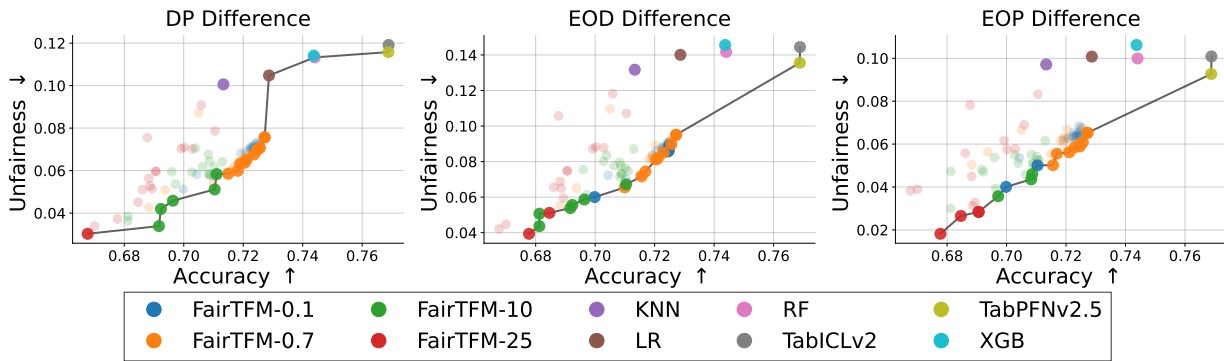

*Figure 2.* Pareto Front between accuracy and fairness for various models. ↑ indicates higher is better (accuracy) and ↓ indicates smaller is better (unfairness).

output token is mapped to class logits through a multi-layer perceptron (MLP).

Our fairness-aware extension is shown in Figure 1b. Rather than operating only on pairs $(x_i, y_i)$, the model processes triplets $(x_i, y_i, s_i)$, where $s_i$ denotes the sensitive attribute. To discourage the learned representation from encoding sensitive information, we attach a secondary prediction head to the query token for predicting the sensitive attribute. This head is preceded by a gradient reversal layer (GRL), which encourages the shared representation to remain predictive of the class label while becoming invariant to the sensitive attribute. Intuitively, this pushes the transformer to reduce reliance on rows and features that are informative for $s$.

To avoid leaking sensitive attributes at inference time, we mask the sensitive attribute in the query using a fixed learnable parameter during both training and inference.

### 3.3. Training Objective

The transformer produces an embedding that captures row- and feature-wise interactions among the context instances, and the two MLP heads output logits for the query label $(\hat{y}_{qy})$ and corresponding sensitive attribute $(\hat{s}_{qy})$. We optimize the model using the following joint cross-entropy objective:

$$\mathcal{L} = \mathrm{CE}(\hat{y}_{qy}, y_{qy}) + \lambda \cdot \mathrm{CE}(\hat{s}_{qy}, s_{qy}) \qquad (1)$$

Minimizing $\mathcal{L}$ encourages accurate prediction of both the target label and the sensitive attribute. However, because the sensitive-attribute branch includes a GRL, the gradient that reaches the shared encoder is sign-reversed. As a result, the encoder is optimized to support label prediction while making sensitive-attribute prediction more difficult. This adversarial interaction encourages the model to rely on features that are informative for the target task but less predictive of the sensitive attribute.

This objective naturally induces a fairness–accuracy trade-off, especially when the target label $y$ and sensitive attribute $s$ are correlated. The parameter $\lambda$ in Eq. 1 controls the

strength of this trade-off. Algorithm 1 provides an overview of the training steps.

## 4. Results

In this section, we describe the experimental setup, including the evaluation tasks, baseline models, and metrics, before discussing the main empirical results. To facilitate reproducibility, we provide anonymous access to the inference code, pretrained checkpoints, and scripts for reproducing the main results at https://github.com/patrikken/FairTFM-inference.

### 4.1. Experimental Setup

**Datasets** We evaluate our model on 120 fairness tasks derived from the 2018 1-Year American Community Survey dataset (Ding et al., 2021). We generate these tasks by varying three factors: (i) the prediction problem (ACSIncome, ACSPublicCoverage, ACSMobility, ACSEmployment, and ACSTravelTime), (ii) the sensitive attribute (Gender, Age, and Race), and (iii) the US state from which the data is sampled (eight total). Additional details are provided in Appendix B.

**Base Models** We compare our method against standard machine learning baselines, including logistic regression (LR), random forest (RF), $k$-nearest neighbors (KNN), and XGBoost (XGB) (Chen & Guestrin, 2016). We also include recent tabular foundation models, namely TabPFNv2.5 and TabICLv2. Because these models do not incorporate any fairness intervention, they provide strong baselines for assessing the fairness limitations of existing approaches on our benchmark. These models do not recieve $s_{qy}$ at inference.

**Metrics** In addition to accuracy and AUROC, we report three fairness metrics: demographic parity difference (DP), equalized odds difference (EOD), and equal opportunity difference (EOP). Definitions and implementation details are provided in Appendix C. Complete architecture, hyperpa-

rameter, and pretraining details are provided in Appendix D.

## 4.2. Results and Discussion

Figure 2 shows our main results: the Pareto frontier between accuracy and fairness for various models. For our FairTFM, we pretrain variants with $\lambda \in \{0.1, 0.7, 10, 25\}$ to control the fairness–accuracy trade-off. TabICLv2 and TabPFNv2.5 achieve the highest average accuracy across tasks, but they also occupy the most unfair region of the Pareto frontier, suggesting that their predictive gains come with a substantial fairness cost. In contrast, the classical machine learning baselines are generally less accurate, yet they remain similarly unfair. Our FairTFM variants consistently improve fairness metrics while maintaining competitive—and often stronger—accuracy than classical baselines such as LR and KNN.

The results also highlight the role of $\lambda$ in shaping this trade-off. Larger values of $\lambda$ (e.g., $\lambda = 25$) move the model toward the fairer region of the Pareto frontier, often yielding more Pareto-dominant solutions there, but at the expense of some predictive accuracy. Smaller values of $\lambda$, by contrast, prioritize accuracy and therefore place the model in less fair regions of the frontier. These trends indicate that our training objective provides a simple and effective mechanism for navigating different operating points depending on the fairness requirements of the application. We observe the same trend when measuring predictive performance with respect to AUCROC. Figure 4 in the Appendix shows the corresponding Pareto frontier. Moreover, Table 1 in the Appendix complements Figure 2 by reporting the corresponding aggregated metrics over all 120 tasks, including standard deviations across three random seeds. Whereas Figure 2 emphasizes the Pareto frontier traced out by different checkpoints and fairness weights, Table 1 makes the overall pattern explicit at the level of final checkpoints: the strongest unconstrained TFMs achieve the highest accuracy but are also the least fair, while FairTFM variants deliver substantial reductions in DP, EOD, and EOP with only a moderate loss in predictive performance.

**Performance dynamics during training** Figure 3 provides a finer-grained view of how fairness emerges over the course of pretraining. We track accuracy and fairness (DP and EOD) for FairTFM with $\lambda \in \{0.7, 25\}$ and for nanoTabPFN, evaluating checkpoints throughout training and averaging the resulting curves over 10 random real-world fairness tasks from our benchmark. The shaded regions indicate variability across these tasks. Since lower DP and EOD correspond to fairer behavior, the middle and right panels show that both FairTFM variants achieve a fairness advantage over nanoTabPFN very early in training and retain this advantage throughout optimization. This gap is not confined to a narrow set of checkpoints: it persists across most of the

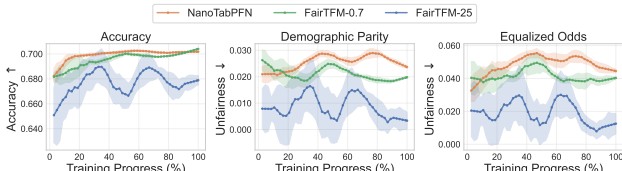

*Figure 3.* Training dynamics comparison between nanoTabPFN (without fairness constraint) and FairTFM with $\lambda \in \{0.7, 25\}$.

training trajectory and becomes especially pronounced in the later stages.

At the same time, the figure makes the fairness–accuracy trade-off induced by $\lambda$ visually explicit. Specifically, nanoTabPFN and FairTFM with $\lambda = 0.7$ converge to very similar final accuracies, but the latter does so while maintaining consistently lower DP and EOD gaps. By contrast, FairTFM with $\lambda = 25$ occupies a distinctly different operating regime: its accuracy remains below that of the other two models, but it attains by far the lowest unfairness on both metrics. The trajectories are also mildly non-monotonic, which is expected because fairness and accuracy are measured on heterogeneous downstream tasks rather than on the pretraining objective itself. Taken together, these dynamics show that our objective does not merely identify a fair endpoint after training; it changes the optimization path itself, steering the encoder toward representations that are progressively less informative about the sensitive attribute, with $\lambda$ providing a direct handle on the final fairness–accuracy operating point.

We report additional results in the appendix. Appendix E.1 shows that FairTFM remains competitive against task-specific models trained with fairness constraints, while Appendix E.2 shows that its learned embeddings also transfer well to downstream fair classification, often yielding favorable fairness–accuracy trade-offs relative to standard fair preprocessing methods. Taken together, these results suggest that FairTFM is not only effective as an end-to-end fair predictor, but also useful as a general-purpose source of fairness-aware representations for downstream models.

## 5. Conclusion

We presented a fairness-aware pretraining framework for tabular foundation models that preserves the single-pass inference setting of ICL. The key idea is to synthesize fairness tasks during pretraining and couple label prediction with adversarial sensitive-attribute prediction through a gradient reversal layer. On 120 ACS-based fairness tasks, the resulting model consistently improves demographic parity and equalized odds while maintaining competitive accuracy. These results show that fairness can be built into TFMs at pre-training time rather than introduced only through task-specific post-hoc correction.

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

# A. Additional Related Work

**Fairness interventions.** Classical fairness-aware learning methods are often grouped by where the intervention occurs in the pipeline (Cresswell, 2025). Preprocessing methods attempt to reduce bias directly in the data, for example through label massaging (Kamiran & Calders, 2009) or fair representation learning (Zemel et al., 2013). In-processing methods instead modify the training objective to encourage fairer behavior during optimization (Zhang et al., 2018). Post-processing methods operate on a trained model's outputs to enforce fairness constraints after training (Hardt et al., 2016). These families of methods have been influential, but they typically assume task-specific model access, retraining, or output recalibration.

**Fairness in in-context learning.** These assumptions become restrictive for foundation models used through ICL, where predictions are generated without task-specific parameter updates. In our setting, the goal is not to retrofit fairness onto a frozen predictor after deployment, but to pretrain a model whose internal representations already support fairer predictions at inference time. Our method therefore combines aspects of in-processing and representation learning: fairness is encouraged during training, yet inference remains a single forward pass without additional optimization.

**Tabular foundation models.** Work on deep learning for tabular data has evolved from specialized supervised architectures (Gorishniy et al., 2021) to pretrained tabular foundation models that generalize across tasks through ICL. TabPFN (Hollmann et al., 2023) established this direction by showing that transformers trained on synthetic supervised tasks can perform strong zero-shot prediction. Subsequent models, including TabPFNv2.5 (Grinsztajn et al., 2026) and TabICLv2 (Qu et al., 2026), improved efficiency and scale, while models such as TabDPT (Ma et al., 2025) and related real-data pretraining approaches (Garg et al., 2025) sought to better align training with real-world tabular distributions.

**Fairness-aware TFMs.** Among prior TFM work, FairPFN (Robertson et al., 2025) is the closest to our setting because it also incorporates fairness during pretraining. However, FairPFN is designed around a causal notion of fairness, using structurally generated datasets with biased and fair outcomes to target counterfactual fairness (Kusner et al., 2017). Our focus is different: we study statistical group fairness notions, namely demographic parity, equal opportunity, and equalized odds, which are more directly aligned with the evaluation metrics commonly used in fair classification benchmarks.

# B. Datasets

In this section, we provide more details about the datasets used for evaluation. We describe the prediction tasks and the dataset construction.

## B.1. Prediction Task Details

We construct our benchmark from tasks provided by the `folktables` benchmark (Ding et al., 2021), which is derived from the American Community Survey (ACS) Public Use Microdata Sample (PUMS). In particular, we consider five binary prediction tasks commonly used in prior work for fairness evaluation:

- **ACSIncome**: predict whether an individual's annual income exceeds $50,000. Following the standard task definition, we restrict the data to individuals older than 16 who worked at least one hour per week during the previous year and earned at least $100.
- **ACSMobility**: predict whether an individual lived at the same address one year earlier. We focus on individuals between 18 and 35 years old, which makes the task less imbalanced than in the full population, where most individuals do not move within a year.
- **ACSTravelTime**: predict whether an individual's commute exceeds 20 minutes. The task is defined on employed individuals older than 16, and the 20-minute threshold roughly matches the median commute time in the 2018 ACS PUMS data.
- **ACSEmployment**: predict whether an individual is employed. For this task, we consider individuals between 16 and 90 years old.
- **ACSPublicCoverage**: predict whether an individual receives public health insurance coverage. We restrict the sample to individuals younger than 65 with income below $30,000, thereby focusing on lower-income individuals who are not eligible for Medicare.

## B.2. Task Construction

Generating new learning signal from randomly selected features in pretraining data is a core concept in self-supervised learning (Sui et al., 2024). In our work, we specifically construct fairness tasks, which is necessary to scale up pretraining, due to insufficient existing datasets labeled with fairness attributes. For each base prediction task, we instantiate fairness evaluation settings by combining it with three sensitive attributes—Gender, Age, and Race—and with data drawn from eight states: Alabama (AL), California (CA), Hawaii (HI), Indiana (IN), Maine (ME), Michigan (MI), New Mexico (NM), and New York (NY). We choose these states to span a range of bias levels reported in the original `folktables` study (Ding et al., 2021), so that the benchmark includes settings with meaningfully different fairness profiles rather than a narrow slice of the ACS. This yields a total of $5 \times 3 \times 8 = 120$ tasks. In other words, each prediction problem contributes 24 tasks, each sensitive attribute appears in 40 tasks, and each state contributes 15 tasks to the full benchmark. For the sensitive attributes, Age is binarized using a 25-year-old threshold, and Race is restricted to White and Black Americans. Across all task instantiations, dataset sizes vary from roughly 3.5k to 12k samples.

This evaluation design is important for two reasons. First, varying the prediction problem, sensitive attribute, and state allows us to assess fairness across substantially different label distributions, demographic compositions, and regional contexts, rather than tailoring conclusions to a single task configuration. Second, reporting results over the full Cartesian product reduces the risk that observed fairness improvements are driven by a small number of favorable settings. We therefore view performance aggregated over these 120 tasks as a stronger indicator of whether a method learns fairness-aware behavior that transfers across heterogeneous real-world tabular prediction problems.

## C. Fairness metrics

In this work, we focus on group fairness criteria that quantify disparities in model behavior across demographic groups. Let $\hat{Y} = f(X)$ denote the binary prediction of a classifier, let $Y \in \{0, 1\}$ be the ground-truth label, and let $S \in \{0, 1\}$ denote the sensitive attribute. We consider the following three standard fairness notions.

- **Demographic parity (DP)** requires the rate of positive predictions to be the same across groups (Dwork et al., 2012). Formally,
$$\mathbb{P}(\hat{Y} = 1 \mid S = 0) = \mathbb{P}(\hat{Y} = 1 \mid S = 1). \tag{2}$$

- **Equalized odds (EOD)** requires the predictor to have the same true positive and false positive rates across groups (Hardt et al., 2016). Equivalently, for each label value $y \in \{0, 1\}$,
$$\mathbb{P}(\hat{Y} = 1 \mid S = 0, Y = y) = \mathbb{P}(\hat{Y} = 1 \mid S = 1, Y = y). \tag{3}$$

- **Equal opportunity (EOP)** focuses only on parity of true positive rates across groups. It can be viewed as the $y = 1$ special case of equalized odds:
$$\mathbb{P}(\hat{Y} = 1 \mid S = 0, Y = 1) = \mathbb{P}(\hat{Y} = 1 \mid S = 1, Y = 1). \tag{4}$$

In the experiments, we report empirical disparity versions of these metrics. For demographic parity, we use the absolute difference in the expected positive prediction rate across groups:

$$\mathrm{DP} = \left| \mathop{\mathbb{E}}_{x \mid S=0} \left[ \mathbb{I}\{\hat{Y} = 1\} \right] - \mathop{\mathbb{E}}_{x \mid S=1} \left[ \mathbb{I}\{\hat{Y} = 1\} \right] \right|. \tag{5}$$

Where $\mathbb{I}(\cdot)$ denotes the indicator function.

For the equalized-odds-based metrics, we define the group gaps in false positive rate and true positive rate using the same expectation notation:

$$\delta_{\mathrm{FPR}} = \left| \mathop{\mathbb{E}}_{x \mid S=0, Y=0} \left[ \mathbb{I}\{\hat{Y} = 1\} \right] - \mathop{\mathbb{E}}_{x \mid S=1, Y=0} \left[ \mathbb{I}\{\hat{Y} = 1\} \right] \right|, \tag{6}$$

and

$$\delta_{\mathrm{TPR}} = \left| \mathop{\mathbb{E}}_{x \mid S=0, Y=1} \left[ \mathbb{I}\{\hat{Y} = 1\} \right] - \mathop{\mathbb{E}}_{x \mid S=1, Y=1} \left[ \mathbb{I}\{\hat{Y} = 1\} \right] \right|. \tag{7}$$

We then report

$$\mathrm{EOD} = \max\left( \delta_{\mathrm{FPR}}, \delta_{\mathrm{TPR}} \right) \tag{8}$$

and

$$\text{EOP} = \delta_{\text{TPR}}. \tag{9}$$

Smaller values of DP, EOD, and EOP indicate fairer behavior, with zero corresponding to perfect parity under the respective criterion. We use these empirical gaps because they provide an interpretable summary of group-level disparities and are standard in fairness evaluations for binary classification.

## D. Model Architecture and Hyperparameters

Our model builds on the nanoTabPFN architecture from `TFM-Playground` [1], which we use as a lightweight transformer backbone for fairness-aware pretraining. Concretely, we use a model with 6 transformer layers, 6 attention heads, embedding dimension 192, and feed-forward hidden dimension 192. Consistent with the main architecture described in Section 3, we augment this backbone with three input encoders—for features, targets, and sensitive attributes—that map their respective inputs into the shared 192-dimensional token space. The label-prediction head and the sensitive-attribute head are both implemented as two-layer MLPs with hidden size 768.

**Pretraining setup.** We pretrain on 300,000 synthetically generated tabular datasets sampled from the TabICL's prior implementation (Qu et al., 2026). Each sampled task contains 150 datapoints, 6 features, and 2 classes, and training is performed with batch size 32. During pretraining, the model receives context and query sets as described in Section 3, with the query label masked and the query sensitive attribute replaced by the learned mask token. All reported FairTFM results are obtained from checkpoints trained under this shared setup, varying only the fairness weight $\lambda$ in the joint objective.

**Optimization details.** We optimize the model using Schedule-Free AdamW (Defazio et al., 2024; Loshchilov & Hutter, 2019) with learning rate $1 \times 10^{-4}$ and no weight decay. This choice provided stable optimization across the fairness weights considered in the main experiments. For the fairness-aware variants, the only task-level hyperparameter we vary is $\lambda$, which directly controls the strength of the adversarial sensitive-attribute objective and thereby the fairness–accuracy trade-off.

**Pretraining details** As summarized in Algorithm 1, each pretraining step instantiates a new fairness task, forms context and query sets for ICL, masks the query label and sensitive attribute to preserve the intended inference setting, and then updates the shared transformer encoder through the coupled label-prediction and sensitive-attribute losses.

---

**Algorithm 1** FairTFM pretraining

---

**Require:** Data prior $p(\mathcal{D})$, FairTFM parameters $\theta$, fairness weight $\lambda$
1: **while** not converged **do**
2:      Sample a dataset $(X, y) \sim p(\mathcal{D})$
3:      Randomly choose one feature of $X$ as the sensitive attribute $s$
4:      **if** $s$ is continuous **then**
5:          Discretize $s$ into categorical groups via "stick-breaking" quantile bins
6:      **end if**
7:      Split $(X, y, s)$ into context $(X_{\text{ctx}}, y_{\text{ctx}}, s_{\text{ctx}})$ and query $(X_{\text{qy}}, y_{\text{qy}}, s_{\text{qy}})$
8:      Mask $y_{\text{qy}}$ and replace $s_{\text{qy}}$ with the learned sensitive-mask.
9:      Get $d$-dimensional vectors $x_i$, $y_i$, and $s_i$ using their respective MLP encoders.
10:     Encode the $d$-dimensional triplets $(x_i, y_i, s_i)$ with the shared transformer encoder
11:     Predict query labels $\hat{y}_{\text{qy}}$ with the main head
12:     Predict query sensitive attributes $\hat{s}_{\text{qy}}$ with the adversarial head through a GRL
13:     Compute $\mathcal{L} = \text{CE}(\hat{y}_{\text{qy}}, y_{\text{qy}}) + \lambda \, \text{CE}(\hat{s}_{\text{qy}}, s_{\text{qy}})$
14:     Update $\theta$ by backpropagation; the GRL reverses gradients from the sensitive-attribute head into the encoder
15: **end while**

---

*Table 1.* Average accuracy and fairness metrics across the 120 benchmark tasks, reported as mean $\pm$ standard deviation over three random seeds. This table complements Figure 2 by summarizing the final-checkpoint performance of each model family. The same trend is visible in tabular form: recent unconstrained TFMs attain the highest average accuracy but also the largest fairness gaps, classical baselines are generally less accurate without being substantially fairer, and FairTFM variants provide better fairness–accuracy trade-offs, with larger $\lambda$ yielding progressively lower unfairness at the cost of reduced accuracy.

| Model | Accuracy | DP Diff | EOD Diff | EOP Diff |
|---|---|---|---|---|
| TabICLv2 | $\mathbf{0.771 \pm 0.05}$ | $0.120 \pm 0.11$ | $0.140 \pm 0.10$ | $0.096 \pm 0.09$ |
| TabPFNv2.5 | $0.770 \pm 0.05$ | $0.121 \pm 0.12$ | $0.145 \pm 0.11$ | $0.098 \pm 0.10$ |
| XGB | $0.744 \pm 0.06$ | $0.116 \pm 0.11$ | $0.144 \pm 0.11$ | $0.097 \pm 0.10$ |
| KNN | $0.716 \pm 0.07$ | $0.106 \pm 0.11$ | $0.139 \pm 0.10$ | $0.093 \pm 0.09$ |
| RF | $0.745 \pm 0.06$ | $0.113 \pm 0.11$ | $0.147 \pm 0.11$ | $0.100 \pm 0.10$ |
| LR | $0.728 \pm 0.06$ | $0.107 \pm 0.12$ | $0.134 \pm 0.11$ | $0.094 \pm 0.09$ |
| FairTFM-0.7 | $0.729 \pm 0.07$ | $0.077 \pm 0.13$ | $0.098 \pm 0.13$ | $0.065 \pm 0.10$ |
| FairTFM-1.0 | $0.726 \pm 0.07$ | $0.075 \pm 0.12$ | $0.091 \pm 0.12$ | $0.064 \pm 0.09$ |
| FairTFM-10 | $0.711 \pm 0.07$ | $0.063 \pm 0.12$ | $0.076 \pm 0.12$ | $0.051 \pm 0.08$ |
| FairTFM-25 | $0.688 \pm 0.06$ | $\mathbf{0.059 \pm 0.14}$ | $\mathbf{0.072 \pm 0.15}$ | $\mathbf{0.025 \pm 0.06}$ |

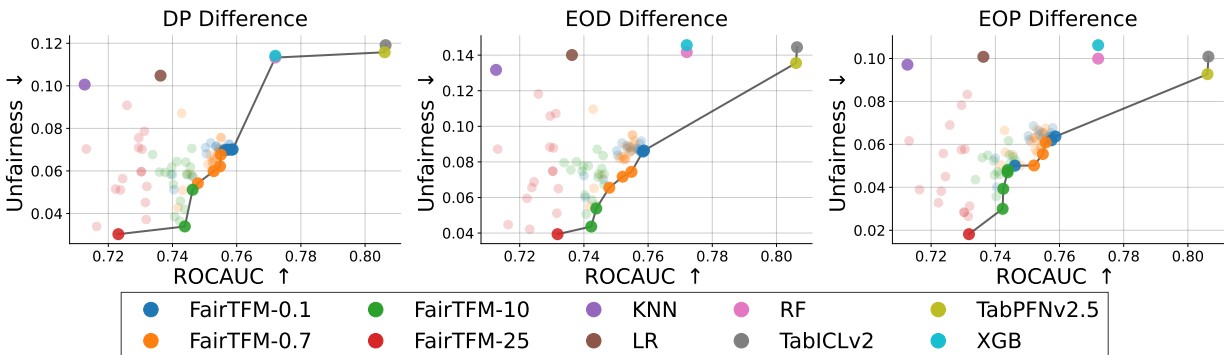

*Figure 4.* Pareto Front between AUCROC and fairness for various models. $\uparrow$ indicates higher is better (accuracy) and $\downarrow$ indicates smaller is better (unfairness).

# E. Additional Results

## E.1. Comparing FairTFM against classical fairness-aware models.

The results above show that FairTFM improves fairness substantially while remaining competitive with standard classical baselines such as LR and KNN. A stronger comparison, however, is against classical methods that are themselves explicitly optimized for fairness. To this end, we evaluate three tabular models—LR, RF, and XGB—augmented with the Exponentiated Gradient (EG) reduction of Agarwal et al. (2018), which enforces group-fairness constraints during training. We focus on ACSIncome in Alabama (AL) and instantiate three fairness tasks by varying the sensitive attribute over gender, race, and age. For each task, we use a random 80/20 train–test split and sweep the EG fairness-violation tolerance over $[0.01, 0.02, \ldots, 0.1, 0.2, \ldots, 1.0]$, with finer resolution in the low-violation regime to better characterize the high-fairness end of the trade-off. We average the results across three random seeds. This setup allows us to compare FairTFM not only to strong predictive baselines, but also to established in-processing fairness methods under a matched downstream evaluation protocol.

---

[1]https://github.com/automl/TFM-Playground

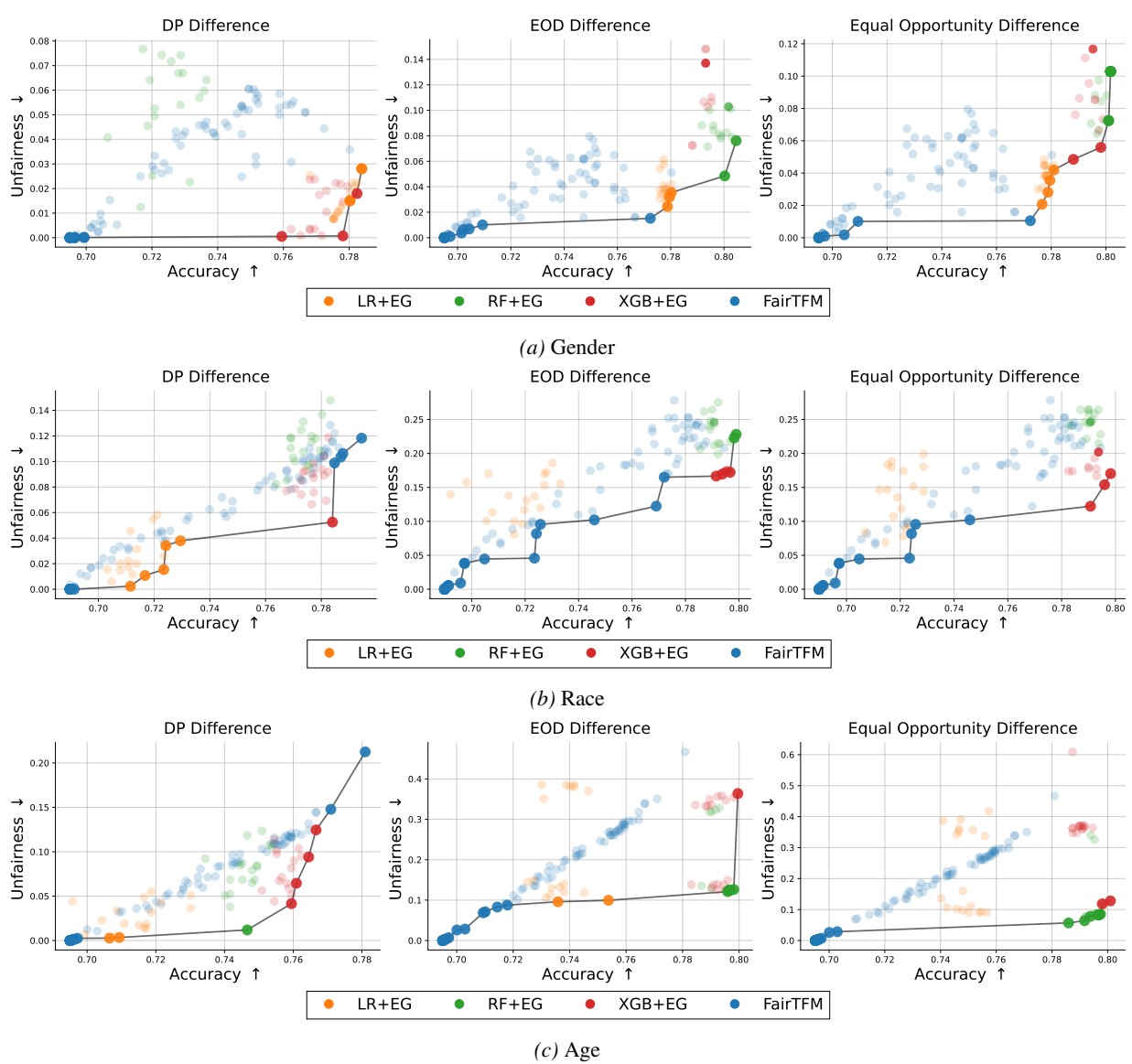

*(a)* Gender

*(b)* Race

*(c)* Age

*Figure 5.* Fairness-accuracy tradeoff comparison with task-specific classical models trained with fairness constraints using Exponentiated Gradient (EG).

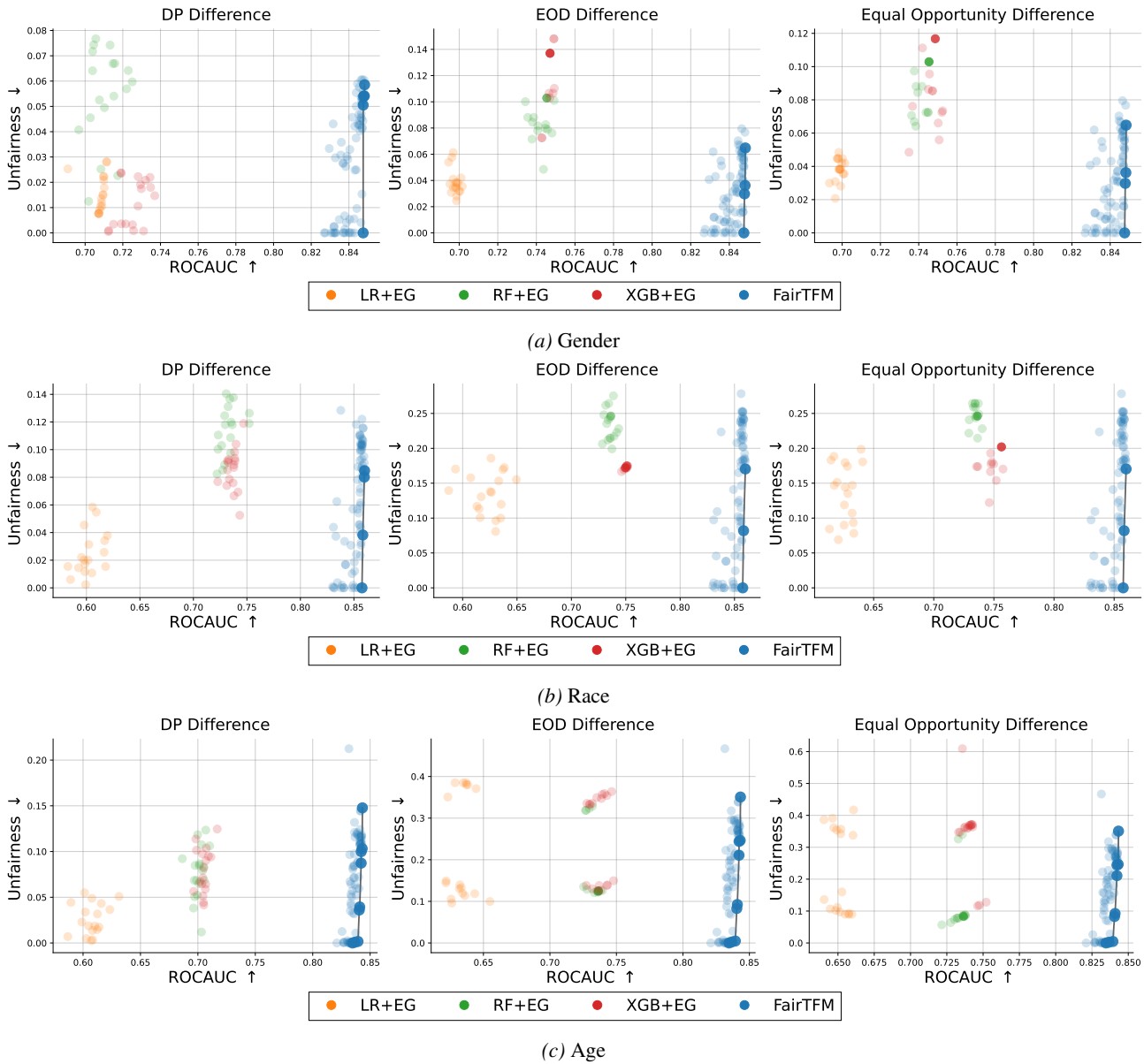

*Figure 6.* Pareto front of classical models trained with fairness constraints using Exponentiated Gradient (EG). FairTFM show best overall Pareto front when performance is measured with ROCAUC.

Figure 5 shows that FairTFM remains highly competitive even against these fairness-constrained baselines. Across all three sensitive attributes, its checkpoints trace out a broad Pareto frontier, indicating that a single pretrained model can realize multiple fairness–accuracy operating points without retraining. For gender, FairTFM attains especially strong trade-offs for EOD and EOP, matching or improving upon the frontier formed by EG-based baselines while remaining competitive on DP. For race, where all methods incur larger fairness gaps, FairTFM still spans a wide and competitive portion of the frontier, particularly at the lower-unfairness end. For age, the comparison is more metric-dependent: EG-based classical models achieve stronger accuracy–fairness trade-offs for EOD and EOP, whereas FairTFM remains competitive on DP but exhibits a clearer trade-off between predictive performance and these stricter parity criteria. Overall, the main advantage of FairTFM is not that it dominates every baseline on every metric, but that it delivers competitive Pareto-efficient solutions across heterogeneous fairness tasks in a single forward pass, whereas the classical alternatives must be retrained with task-specific fairness constraints for each new setting. Moreover, as shown in Figure 6, FairTFM provides the best overall Pareto front when predictive performance is measured through ROCAUC.

## E.2. FairTFM as a Fair Representation Learner

We hypothesize that the encoder learned by FairTFM produces representations that are useful for downstream fair prediction. We validate this hypothesis empirically.

Fair representation learning is a well-studied direction in fair machine learning. These methods learn a transformation of the input that suppresses information about the sensitive attribute while preserving the signal relevant to the task. More formally, we consider the following notion:

**Definition E.1** (Fair Representation). Let $(X, Y, S)$ be random variables where $X \in \mathbb{R}^{m \times d}$ is the input, $Y \in \mathcal{Y}$ is the target label, and $S \in \mathcal{S}$ is a sensitive attribute. Let $g : \mathbb{R}^{m \times d} \to \mathbb{R}^{m \times d'}$ be a representation map and define $Z := g(X)$. We say that $Z$ is a fair representation (with respect to $S$) if

$$Z \perp\!\!\!\perp S \mid Y.$$

The above condition is equivalent to $\mathbb{P}(Z \mid Y, S) = \mathbb{P}(Z \mid Y)$ or, in information-theoretic terms, $I(Z; S \mid Y) = 0$. A downstream model trained on $Z$ is therefore expected to exhibit improved fairness relative to a model trained directly on $X$.

Definition E.1 is conceptually aligned with the objective of FairTFM, which encourages the learned query embedding to suppress sensitive-attribute information while preserving task-relevant signal.

Existing approaches for learning $g$ are typically task-specific, requiring a separate model to be fitted for each downstream task (Zemel et al., 2013; Madras et al., 2018). We compare FairTFM against two open-source, sklearn-compatible preprocessing methods from the `fairlearn` library (Bird et al., 2020). The first is Correlation Remover (CR), which reduces linear dependence on the sensitive attribute by applying a linear transformation to the non-sensitive features (Bird et al., 2020). The second is Learning Fair Representations (LFR), which maps inputs to latent prototypes while encouraging similar assignment behavior across demographic groups (Zemel et al., 2013).

We consider ACSIncome for Alabama (AL) with three sensitive attributes—gender, race, and age—yielding three downstream fairness tasks. Each task is randomly split into 80% training data and 20% test data. For CR and LFR, the training split is used to fit the representation map $g$, which is then applied to both the training and test features. For FairTFM, we use the training split as context and compute embeddings for both training and test examples from the transformer's output. Importantly, these representations are obtained in a single forward pass, without any task-specific gradient updates.

We train three downstream classifiers—LR, RF, and XGB—on four types of representations: the raw input features and the fair representations produced by CR, LFR, and FairTFM. For CR and LFR, we sweep the parameter controlling the fairness–accuracy trade-off over $[0.01, 0.02, \ldots, 0.1, 0.2, \ldots, 1.0]$, with denser coverage in the higher-fairness regime. For FairTFM, we evaluate the checkpoints associated with the $\lambda$ values used in Figure 2. We run this experiment across three random seeds and average the results.

Figure 7 showcases the Pareto front of XGBoost models trained using data representation from different fair methods. As can be seen, FairTFM can provide competitive Pareto points on downstream tasks across sensitive attributes. The results show the ability of our model to generate embeddings that encode less information about the given sensitive attributes. The downstream models trained using our embedding achieve similar fairness-accuracy performance relative to task-specific methods. Our methods stand out due to their ability to adapt to new tasks via ICL without needing task-specific optimization. Figure 8 and 9 find a similar trend with logistic regression and random forest as downstream models respectively. Correlation remover is more effective on LR model, since it only removes linear correlation of the sensitive attributes, and can still exhibit higher unfairness when applied to non-linear models.

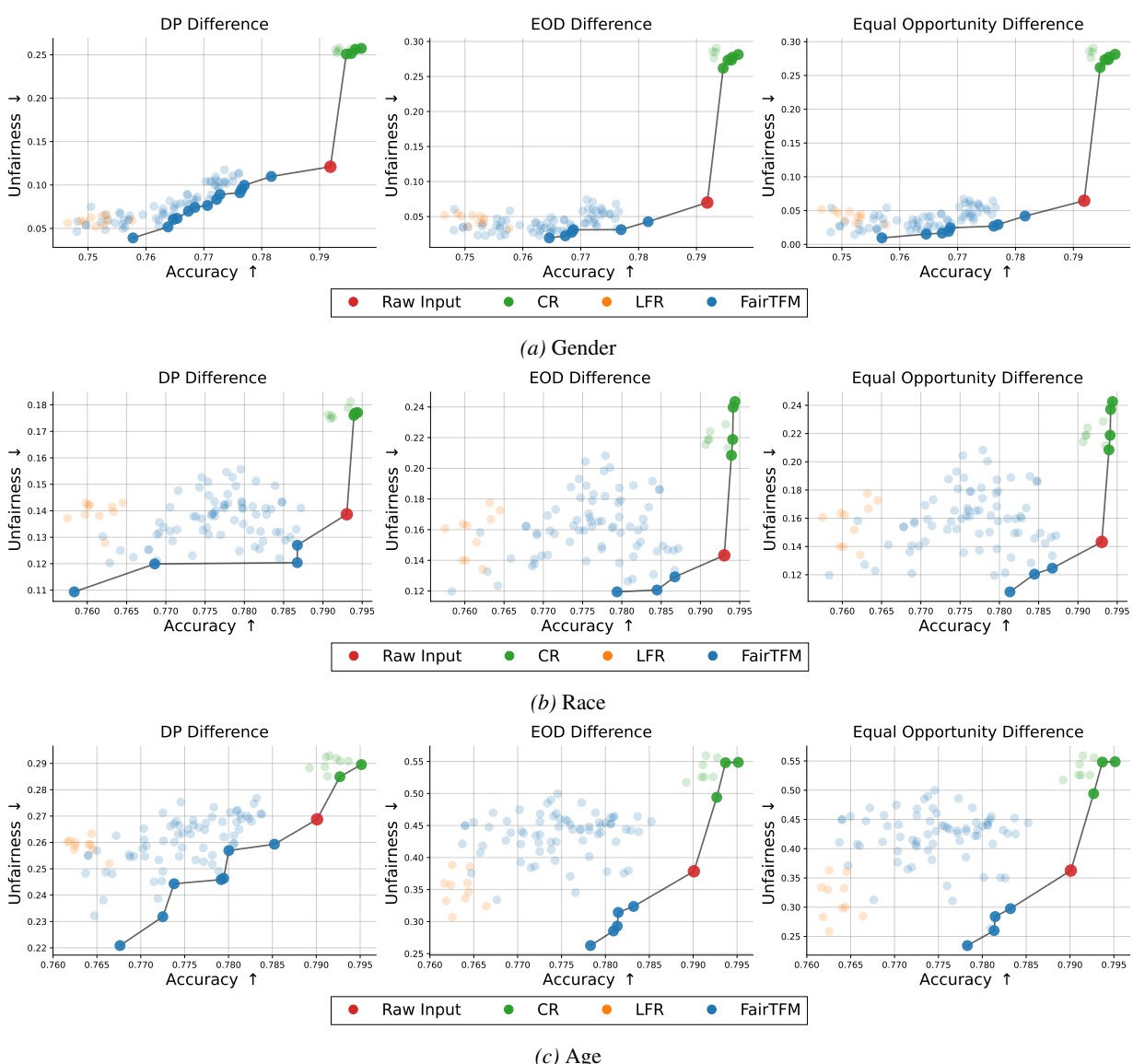

*Figure 7.* Pareto front of XGBoost models trained with different data representations.

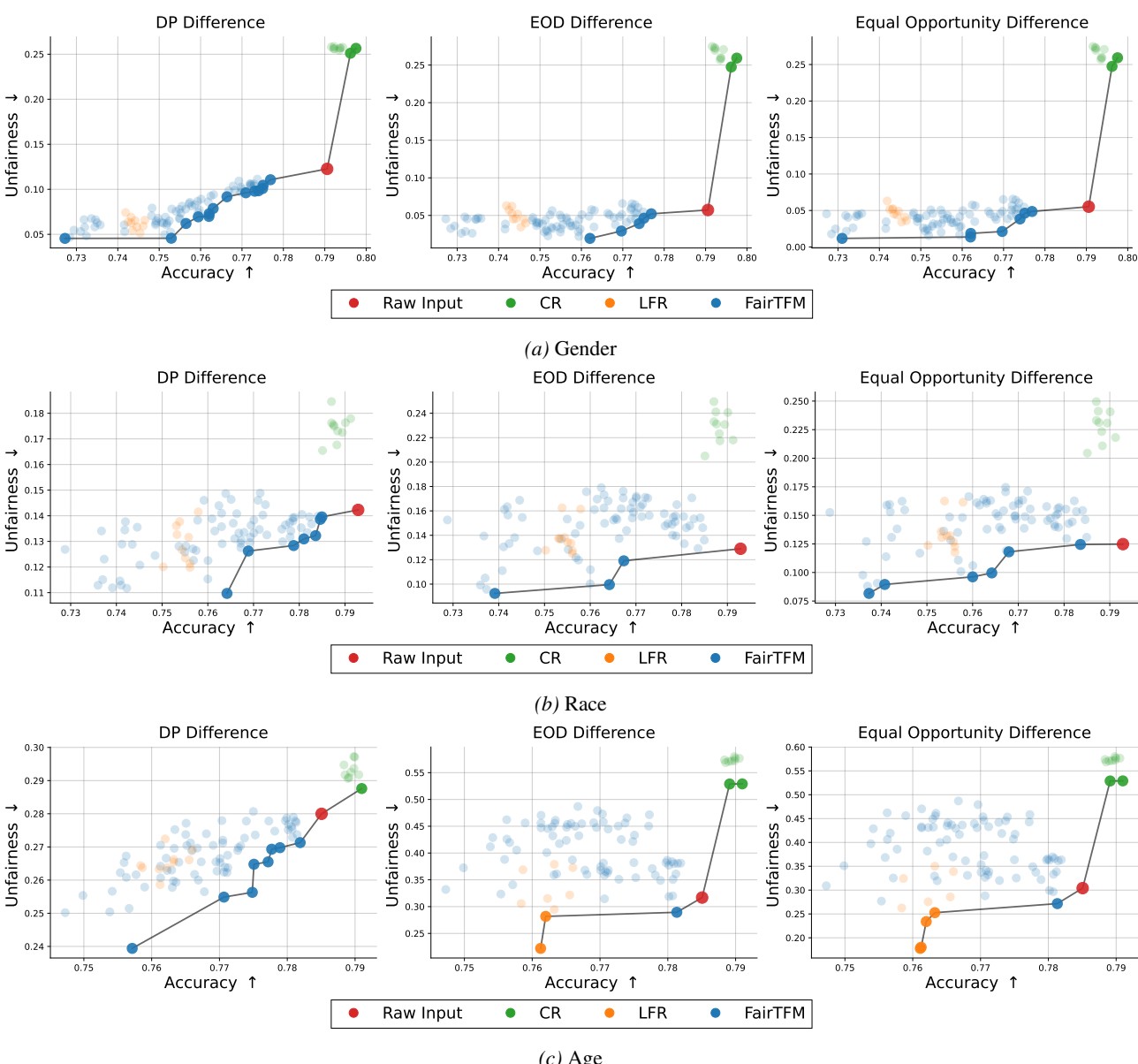

*(a)* Gender

*(b)* Race

*(c)* Age

*Figure 8.* Pareto front of Random Forest models trained with different data representations.

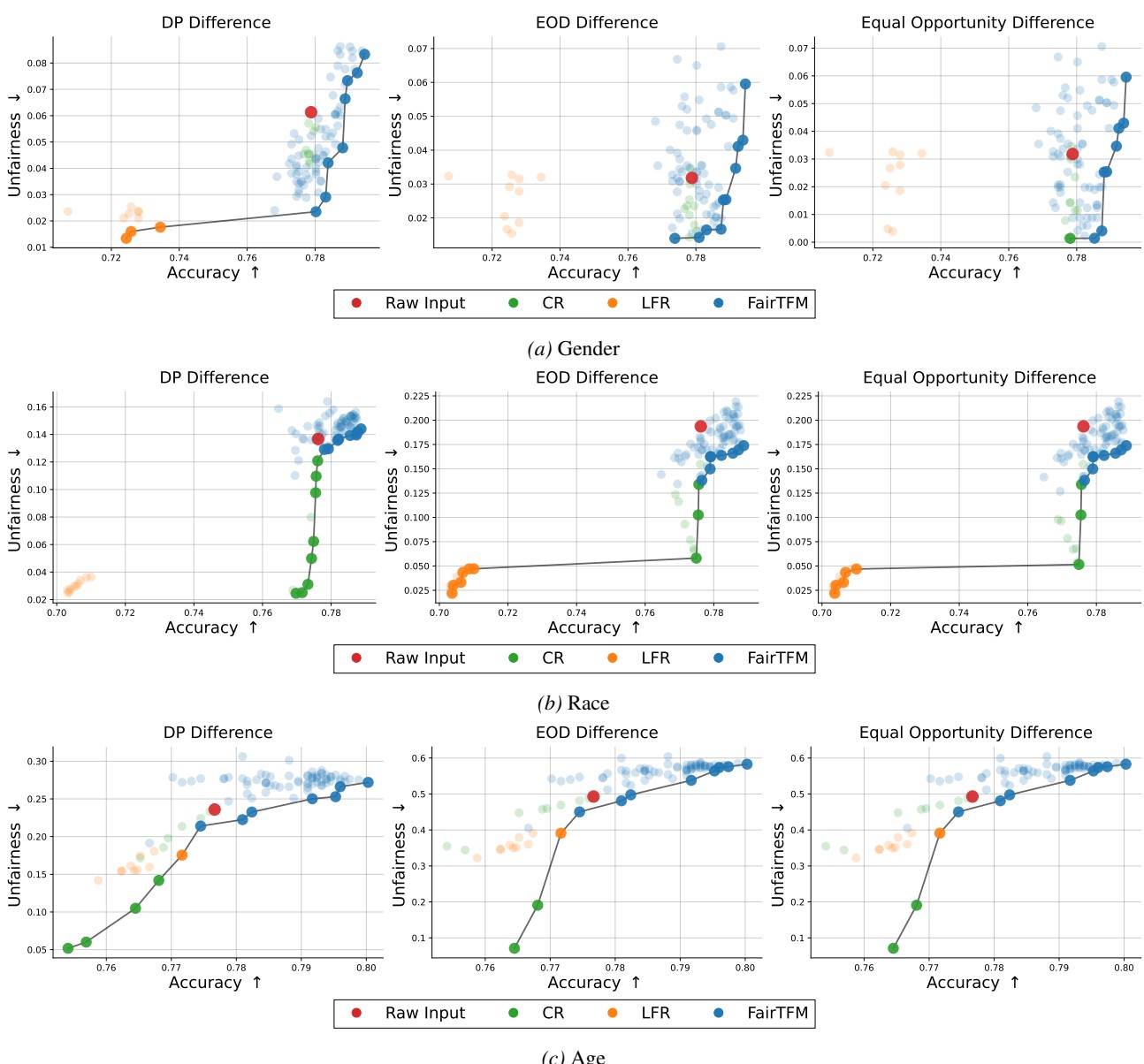

*(a)* Gender

*(b)* Race

*(c)* Age

*Figure 9.* Pareto front of Logistic Regression models trained with different data representations.

### E.3. Evaluation beyond ACS PUMS tasks.

The evaluation in Section 4 is based on 120 tasks derived from the same source, ACS PUMS (Ding et al., 2021). To assess whether the observed fairness–utility trends extend beyond this data source, we additionally evaluate on a collection of widely used fairness benchmarks spanning different domains, prediction targets, and sensitive attributes. Specifically, we consider the following datasets:

- The Adult (Census Income) dataset contains demographic and socioeconomic records from the U.S. Census (Becker & Kohavi, 1996). The task is to predict whether an individual's annual income exceeds $50,000, using gender, race, or age as the sensitive attribute.

- The COMPAS dataset contains criminal justice screening records for defendants (Angwin et al., 2022). The task is to predict whether an individual will be rearrested within two years of their initial arrest, with race as the sensitive attribute.

- The German Credit dataset contains records of bank account holders and is commonly used for credit risk assessment. The task is to classify applicants as low-risk (good credit) or high-risk (bad credit), using sex or age ($\leq 25$ years) as the sensitive attribute.

- The Diabetes dataset contains clinical records from 130 U.S. hospitals and integrated delivery networks collected between 1999 and 2008 (Strack et al., 2014). The task is to predict whether a patient will be readmitted within 30 days of discharge, using gender or race as the sensitive attribute.

- The Law School dataset consists of admissions records collected by the Law School Admission Council (LSAC) from 163 U.S. law schools in 1991 (Wightman, 1998). The task is to predict whether a candidate will pass the bar exam, using race or gender as the sensitive attribute.

- The CelebA dataset provides facial-attribute annotations for celebrity images. We consider two binary prediction tasks, *Blond Hair* and *Smiling*, using gender as the sensitive attribute in both cases.

Together, these benchmarks define 12 additional fairness tasks. We use the same evaluation protocol as in Section 4 and report Pareto fronts averaged across tasks. As shown in Figure 10, the qualitative conclusions from ACS PUMS persist on this broader benchmark suite: FairTFM consistently occupies the low-unfairness region of the Pareto front, whereas TabICLv2 and TabPFNv2.5 achieve the highest predictive performance but incur substantially larger fairness violations. The AUC results in Figure 10b show the same pattern, indicating that FairTFM improves fairness without a commensurate loss in predictive performance. Finally, varying $\lambda$ exposes the expected fairness–utility trade-off: larger values produce fairer FairTFM models, while increasingly prioritizing fairness can reduce predictive accuracy.

## F. Limitations and future work

Despite the strong empirical performance of FairTFM, our study has several limitations. First, our pretraining strategy relies only on TabICL's prior generation, which provides scale and diversity but may not capture all of the semantic and societal structure of real sensitive attributes in downstream deployments. Secon, while FairTFM is broadly competitive, the results also show that it does not dominate specialized fairness-aware baselines on every metric or every task configuration, especially in the stricter settings where age is used as sensitive attribute. We view these limitations not as drawbacks of the overall approach, but as evidence that fairness-aware pretraining opens a rich new research direction. In particular, our findings pave the way for future work on better fairness-task priors, improved trade-off control during pretraining, and foundation models that can adapt their fairness behavior more explicitly to downstream deployment requirements. Finally, we emphasize that our experiments rely on nanoTabPFN, a lightweight backbone chosen to isolate the effect of fairness-aware pretraining. We believe the proposed framework is largely orthogonal to architectural scaling, and integrating it into larger state-of-the-art TFMs such as TabPFNv2.5 or TabICLv2 may yield substantially stronger fairness–accuracy trade-offs through improved representation learning and richer pretrained priors.

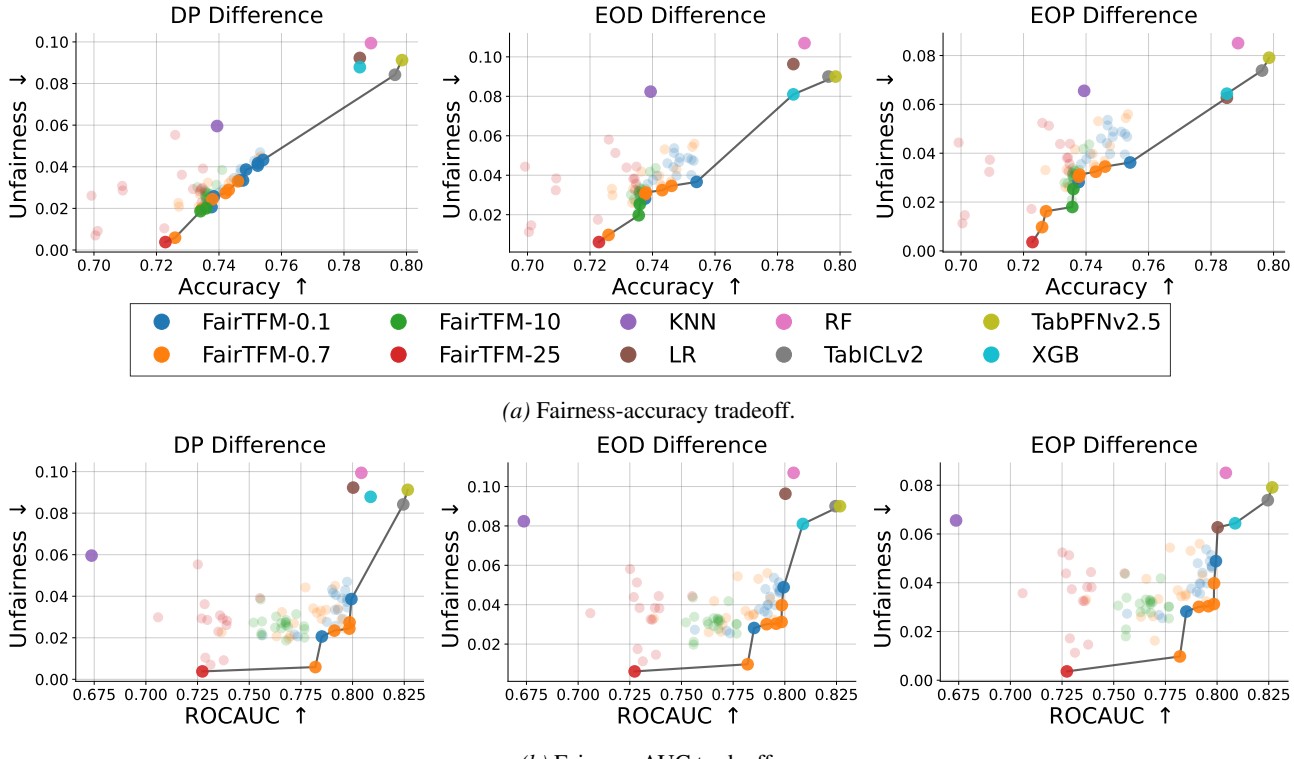

*(a)* Fairness-accuracy tradeoff.

*(b)* Fairness-AUC tradeoff.

*Figure 10.* Pareto Front for various models on 12 fairness tasks beyond ACS PUMS tasks. ↑ indicates higher is better (accuracy) and ↓ indicates smaller is better (unfairness).

