# OpenReview forum: "Training Fair Tabular Foundation Models"
_ICML.cc/2026/Workshop/FMSD — FMSD @ ICML 2026 SpotlightOral_

### Official Review · Reviewer_HEaU · 2026-05-20
**Redesigning tabular foundation model architecture for built-in fairness while not sacrificing accuracy**

**Rating:** 9
**Confidence:** 4

**Review:**

Summary of Contributions
This paper proposes FairTFM, a fairness-aware pretraining framework for Tabular Foundation Models. It introduces a gradient reversal layer architecture to learn representations invariant to sensitive attributes, enabling fair predictions in a single ICL forward pass. Evaluated on 120 ACS-derived (synthetic) fairness tasks, FairTFM consistently improves fairness metrics while maintaining the pareto front on accuracy.

Strengths
- Well-motivated problem — clearly articulates why standard fairness methods fail in the ICL setting.
- Thorough evaluation — 120 tasks (5 problems × 3 attributes × 8 states) with Pareto frontier analysis across DP, EOD, and EOP.
- Clear communication — well-written with effective figures illustrating the trade-off from multiple angles.

Weaknesses
- Minimal; the authors acknowledged but all experiments use the lightweight nanoTabPFN backbone only, and datasets were limited to a single data source (ACS).

Suggestions
- Try on larger TFM architectures (TabPFNv2.5, TabICLv2) that are more SOTA, to show that it's agnostic to base architecture
- Evaluate on additional datasets beyond ACS.
- Explore intersectional fairness with multiple simultaneous sensitive attributes.

---

### Official Review · Reviewer_Q2VH · 2026-05-20
**Pretraining TFMs with fairness contraints, a well-motivated and written paper**

**Rating:** 7
**Confidence:** 3

**Review:**

# Summary
FairTFM incorporates statistical group fairness into TFM pretraining via synthetic fairness task generation (randomly designating input features as sensitive attributes) and an adversarial Gradient Reversal Layer (GRL)-based architecture with a dual prediction head. The result is a TFM that produces more fair predictions in a single ICL forward pass without post-hoc correction. The experiments were carried out on ACS PUMS.

# Strengths
The technical gap is well-motivated as existing fairness interventions are incompatible with ICL's single-pass paradigm, and the GRL-based solution is well-grounded.

Claims are rather proportionate to evidence. The Pareto frontier framing shows the advantage of FairTFM well and reported training dynamics support the claim that fairness gains are persistent.

# Weaknesses
All 120 tasks come from ACS PUMS and generalization to structurally different tabular domains is unverified.

The $\lambda$ parameter controlling the trade-off is not interpretable nor easy to pick. It seems that to understand what $\lambda$ to pick for a given pretraining corpus and expected set of tasks one would have to sweep over a range of $\lambda$ values to attain a desired trade-off.

# Questions
The literature on fairness interventions in tabular data is vast. How does FairTFM compare to task-specific "fair" baselines?